# Reshaping Higher Educational Institutions through Frugal Open Innovation

**Jayamalathi Jayabalan** [1], **Magiswary Dorasamy** [2,*] **and Murali Raman** [3]

1    Accounting Department, Faculty of Accountancy and Management, Universiti Tunku Abdul Rahman, Kajang 43000, Malaysia; jayamalathi@utar.edu.my or 1191400063@student.mmu.edu.my
2    IT Management Department, Faculty of Management, Multimedia University, Cyberjaya 63100, Malaysia
3    Postgraduate & Continuing Education, Asia Pacific University of Technology & Innovation, Kuala Lumpur 57000, Malaysia; murali@apu.edu.my or profdrmuraliraman@gmail.com
*    Correspondence: magiswary.dorasamy@mmu.edu.my

**Abstract:** Many private higher educational institutions (PHEI) are facing poor profitability, increased short term debts with under-resourced cash flow and insufficient funds that could lead to financial distress. To address the issues of ever-changing business environments and to deliver value propositions, PHEI should focus on their intangible assets to increase their capabilities to achieve frugal open innovation. The objective of this paper is to investigate the challenges faced by private universities from the practitioners' points of view and offer a practical solution. This paper also attempts to identify whether there is a need for any changes in business model or operations required by private universities to sustain their competitive advantage in the current environment. This study is exploratory in nature due to scarcity of past literature on frugal open innovation in PHEI context. Interviews were conducted with experienced practitioners to elicit their experience managing challenges in PHEI. As a result, this paper sheds light on the ability of PHEI to formalize, capture, and leverage its intangible assets rather than only investing and managing tangible assets in order to achieve frugal open innovation. Frugal open innovation is the enabler for PHEI to focus on core functions, create closer integration with industry, local and international communities and promote greater efficiency in operations. This paper is novel because it seeks to contribute to the current debate in the literature, positioning frugal open innovation (FOI) within the sphere of intellectual capital research, through exploring the effect of intellectual capital on frugal innovation is mediated through the information technology capability. The result indicates that sales and operating planning (S&OP) can be panacea for the five main challenges faced by PHEI includes structural challenges, operational challenges, financial challenges, social challenges and technological challenges. We conclude that there is a role for intellectual capital to achieve FOI by influencing IT capabilities, thus warrants more research to fill this research gap.

**Keywords:** intellectual capital; frugal open innovation; open innovation dynamics; IT capabilities; higher education institution

## 1. Introduction

Private higher learning institutions (henceforth referred as PHEI) practice knowledge transfer through various collaborative activities such as collaborative research, consultancy, commercialisation activities and spin-off projects. These activities, when done well, can lead to creation and commercialisation of intellectual property, which could generate income for the universities. There is also engagement of human capital between universities and the industries, which is known as academic entrepreneurship. Relational capital of universities that involves collaboration with external parties enable PHEI to gain new insights for research and technological knowledge, which is unachievable through closed innovation. As the education environment becomes more competitive, PHEI need to focus on improving their capabilities due to limited resources. In this regard, open innovation

can provide a platform for PHEI to utilise internal capabilities as well as external collaborations to boost frugal innovation adoption. Open innovation may bring partnerships with external parties that could lower the cost and meet customer requirements to achieve frugal innovation. Nevertheless, successful open innovation also requires external collaboration to source for ideas, technologies, knowledge, and funding. Briefly, for PHEI to succeed with open innovation, PHEI rely upon two vital internal component, which are human capital competencies and structural capital (e.g., system, R&D capabilities and intellectual property). This paper presents an emerging concept called frugal open innovation (FOI) as a solution for PHEI to benefit substantially through networking with external stakeholder to facilitate open innovation practices and its ability to attract government grants, international research collaborations and industry collaborations.

PHEI in Malaysia continues to face institutional decline in teaching and delivering subjects, a lack of quality academic leadership, reductions in funding, and ever increasing cost and technological advancement [1]. Increasingly, local PHEI are going through transformation to boost their effectiveness, efficiency, and transparency to increase competitiveness. Intellectual capital (IC) and frugal open innovation (FOI) together seemingly play a vital role in this regard. According to [2], PHEI produce knowledge through their most valuable resources, which consists of lecturers, researchers, administrative staffs, management team, students and all other stakeholders. PHEI should be able to focus on transferring good knowledge through new methods of learning if they can obtain information from their IC. The intellectual measurements are the main indicator for PHEI to increase their productivity of knowledge-based work. IC is very important not only in enhancing organisational performance but also in strengthening the links between universities and industry to provide academicians as well as practitioners with mutual benefits. Unfortunately, the literatures on the value and contribution of IC to universities and the adequate tools that can be established to generate, manage and evaluate IC is scant [3].

Intellectual capital (IC) is an emerging issue for academics, governments and investors [4]. IC has been investigated in industrialized nations because it has been recognized as the key indicator of an organisational growth in a knowledge economy [5]. However, according to [6], studies remain insufficient in emerging economies and specifically in universities. The measurement and management of IC dimensions are critical. Ref. [7] claimed that, in organisations with knowledge-intensive activities such as PHEI, the variances in the provision of IC can ensure the effectiveness of the processes and the capability of the entities to create value. However, regarding cost and efficiency in performance, the IC value of PHEI is seldom debated [8]. Several calls are made in both academia and industry to seek ways for boosting innovation using alternative approaches to ensure cost effectiveness and meet societal demands without requiring huge investments.

Attention among researchers on frugal innovation is increasing, as it becomes a concept initiated in the context of emerging economies and later practiced in western countries [6]. FOI requires resource efficiency, that is, products and services with less possible cost but with acceptable quality level. FOI is conceptualized as an accessible business model that fulfils the need of customers by shifting resources constraint into value adding through a resource-based view (RBV) theoretical lens and open innovation [9,10]. FOI creates competitive advantages for a firm by producing high-quality and low-cost products; helps the firm optimise resources, competencies and skills to increase its efficiency and reduce wastage [6]. Firms are always challenged with scarce resources, and they continuously look for ways to do more with less. The most challenging issue for research and development (R&D) productivity in an organisation is to efficiently utilize its physical, intellectual and skilled resources using a frugal approach.

Given the above backdrop, the issue of PHEI's sustainability continues to dominate the current debate. Hence, this paper intends to identify the challenges faced by PHEI in Malaysia. This paper also attempts to identify whether there is a need for any changes in business models or operations of PHEI to sustain their competitive advantage in the

current environment with limited resources. Our objective therefore, is to understand the challenges faced by PHEI and provide insights to PHEI—by examining the role of FOI.

This paper proceeds as follows. The first section reviews the literature on the current scenario of PHEI and variables related to FOI, IC, and information technology (IT) capability (ITC). The section continues by explaining the method guiding this research and highlighting the findings obtained. The paper concludes with the researchers' discussion on the overall outcome of the investigation. Hence, the research questions for this study are as follows:

1. What are the challenges faced by PHEI in the current environment?
2. How can PHEI develop resources or capabilities to sustain performance?
3. Is there a need for changes in universities' business models or operations in PHEI to ensure success?
4. How can frugal open innovation play a role in PHEI to face the challenges?

Our findings are derived based on content analysis of small data. Through a several deep dive interview sessions with industry practitioners, using content analysis, the paper provides insights on how FOI can benefit PHEI.

## 2. Literature Review

Recently, the Malaysian PHEI system has experienced a substantial educational transformation—with calls to increase quality of research, comparability, competitiveness, flexibility, and transparency. Yet, an issue in quality education in Malaysian HEI persists despite government efforts. As a result, numerous calls for further research are made to extend the understanding of IC management within HEIs [11] to ensure that HEIs have sufficient intellectual resources to serve their core goals, which include teaching and research [12]. The purpose of the present research is to fill the gap in the existing knowledge on PHEI's performance measurement initiatives, underlining the growing importance of IC.

Poor profitability increased short term debts with under-resourced cash flow, and insufficient fund to pay bills are normally the indicators that many private PHEI are facing financial insolvency or at a high risk of financial failure. Governments are facing serious fiscal stress that require transformation of the higher education system owing to the tremendous increase of higher education and rising unit cost. The budget cut for the public higher education is quite high, from 12% in 2015 and 15% in 2016 to 19% in 2017. The proportion of development spending also dropped from 18.5% per year in 2012–2014 to 16.9% in 2015–2017. The University of Malaya faced a major shortfall in 2016, when it suffered 27% budget reductions from the prior year, followed by the National University of Malaysia's allocation drop by 31% in 2017 [13].

PHEI currently experience an increasing expenditure to achieve the specified level of quality in teaching and research to stay competitive and contribute to the economic growth of the nation. In addition, as reported in one of the leading newspapers in Malaysia, most PHEI are facing financial crisis with 53% suffering losses before tax and 55% suffering losses after tax [14]. The numbers of PHEI suffering losses rose from 41% before 2013 to 55% after 2013. Since 2010, the fall has been up to an average of 54% profit before tax and 78% profit after tax. This tremendous fiscal strain causes various effects in terms of benefits for around 5800 academic staffs and around 121,000 students who are affected by the poor quality of education in loss-making institutions [14]. PHEI needs to pay more focus and attention in adopting potential methods that will create value for its stakeholders through exploiting their internal and external resources and strengthening their capabilities in respond to the dynamic business environment.

Furthermore, there is a growing public demand for better quality improvement for PHEI programmes, especially in their learning outcomes and impact to the community [1]. To overcome the funding crisis and growing resources constraints, PHEI need to focus and pay attention on adopting potential methods that can create value for stakeholders. PHEI need to play an important role in fostering talented and skilled manpower for the

nation to grow towards sustainable economic growth. PHEI not only need to adopt market-oriented strategies, stressing the role of efficiency, economies of scale, rationalization, increase of private contributions or the development of better capabilities to respond to the market [15]. PHEI are also required to adopt strategies to satisfy managerial and academic staff, students, society, industry partnership and "national and international" firm [2,3]. For sustainability, PHEI in emerging economies, which are challenged by constrained resources and flexible improvisations and adopting low-cost approach should seek for newer innovative paradigms such as FOI. In a frugal environment, firms rely on internal and external knowledge and learning capabilities to compensate issues arising due to resource constraint by continuously improving customer satisfaction and profitability for shareholders. Therefore, the aim of this research is to address this substantial issue by determining to what extent IC in PHEI may develop capabilities to meet the criteria identified in FOI.

The world is transforming from industrial economy to an information economy. Recently, the number of studies focusing on 'knowledge conversion process' has been growing [16]. Information technology (IT) is viewed an invisible capability in digitalised era in knowledge economy which can contribute significantly on organisation performance. As mentioned by [17], knowledge is considered as an important resource, thus making IC a preliminary point for exploring this issue. Moreover, ref. [18] examined the relevant uses of information and communication technologies (ICT) with social capital and other knowledge resources and reported that they can enhance firm's performance in a knowledge-intensive business unit.

ICT can increase access and improve human and IC, that is, online databases for recruitment, use of intranets for information and knowledge sharing among staff within firm and idea exchanges with scholars from other institutions. Many prior studies use IT expenditure as a measure of IT value [19]. However, this research focuses on the intangible use of ITC to analyse its influence on innovation and performance. In addition, FOI is used as performance indicator because the intangibles, such as enhanced responsiveness to customers and better coordination with suppliers, are difficult to measure and mostly do not always have direct effect on output, for example, return on investment or return on assets. Undoubtedly, FOI will be beneficial in retaining customers and improving coordination and even help in sustaining profitability. Instead of finding the direct influence of IC on organisation performance, this research aims to find the relationship between IC and ITC that leads to FOI. Firms with better IC management and ITC will lead to superior performance. In sum, this research investigates the relationship between IC, ITC and FOI based on the existing literature.

## 2.1. Integrating Open Innovation and Frugal Innovation

Frugal innovation (FI) means managing the entire value chain with minimal materials and financial resources which improves product quality and reduce cost. Companies can gain benefits through cost reduction owing to limited usage of resources, reuse of existing components, adoption of cost effective technology and simplification of design [20,21]. By engaging in FI, PHEI's adherence to stakeholders' expectations can improve and ultimately lead to profitability and survival. On the marketing point of view, producing resource-saving products and services by focusing on essential function of the product/service and avoiding wasteful spending are crucial in a resource-constrained environment. Ref. [10] defined frugal as 'do more with less' for more people. To date, academics and practitioners have been striving to establish an agreed-upon definition of this concept. Ref. [11] found that the concept of FI overlaps with 'cost, good enough, the base of pyramid, inclusive, grassroots, disruptive, and reverse innovation'. Regardless of the definition, the understanding regarding FI is that the concept is difficult to measure, operationalise and evaluate.

Therefore, various business models have been introduced in an attempt to represent what comprises FI. Refs. [22,23] recommended that an innovation is categorised as frugal only if it fulfils all three criteria which are substantial cost saving, emphasised core functionalities and enhanced performance level. To sustain competitive advantage, firms in emerging economies, which are challenged by constrained resources and flexible improvisations and adopting low-cost approach are seeking for newer innovative paradigms such as FI. An organisation under resource constraint have focus on optimising its internal and external knowledge to develop its learning capabilities in a frugal environment to be able to continuously provide value-added service to its customers and improve financial wealth to its shareholders. FI is observed as core functionality, performance level, usage-centred, ruggedization, cost reduction, no frills strategy and environmental issue [21,22,24,25]. ITC from prior research from dynamic capability theory perspective mediates the correlation between IC and FOI.

As an intellectual based enterprise, PHEI cannot only depending on its internal resources and capabilities but rather incorporate external knowledge and integrate them with their own resources and capabilities to accelerate innovation through new knowledge combinations. Knowledge based open innovation able to support PHEI to exploit opportunity through multiple channels such as intellectual property licensing, joint ventures, spin-offs and leverage their R&D output to external market. Hence, the integration among intellectual capital and open innovation is through an important component of intellectual capital, which is relational capital that increases the opportunities for PHEI to get access of external knowledge and expertise for joint value creation with external stakeholders. Therefore, PHEI should explore its critical dynamic capabilities and equip itself with very strong sensing, seizing and transforming capabilities to develop open innovation approach to foresight opportunities. Horizontal and vertical cooperation with industry, government, business associations, NGOs, research institutions and other universities plays an important role to accelerate the capacity of PHEI to achieve frugal open innovation (FOI).

Open innovation may be critical in boosting frugal innovation in emerging markets. The concept of open innovation plays crucial role for low-cost innovation as it facilitates the flow of information, distribute the risk of R&D costs and increases competition through partnership among various parties. Open innovation includes cyclical dynamics among market open innovation, closed open innovation, and social open innovation from a macrodynamic perspective [26]. Social open innovation, which is spearheaded by social entrepreneurs through interaction between technological and social, which will be the source of market open innovation. Market open innovation focuses on the connections between technology and the market. Finally, market open innovation initiates closed open innovation through collaborations and alliances of large organisations. From microdynamic aspect, if the complexity is well handled, a complex adaptive mechanism can be accomplished through creative creation at the evolutionary change stage. Approaches to open innovation micro-dynamics include going beyond fusion toward IoT through open innovation, business models of information systems and learning mode in the fourth industrial revolution [26]. Besides, open innovation able to incorporate internal and external capital such as mobility of human capital, technological advancement and community engagement that helps organisation to absorbed knowledge resources efficiently to create competitive advantage and new market share. Formerly, closed innovation emphasizes more on intellectual property (IP) during production of value-added product and services. However, open innovation economy focuses more on knowledge sharing through industry convergences, mergers and acquisitions and combining technology with social values [26]. Therefore, via interactions between universities, industry, and government policy, the quadruple-helix model demonstrates creation of knowledge resources by establishing university-industry partnership and engaging proactively with economy, society and environment in open network, government projects, and interactive innovation rather than only serving traditional role and fundamental research [26]. Open innovation able to stimulate

frugal innovation to develop affordable products with cost-constraint by reducing R&D cost through sharing risk with external parties.

PHEI that have the ability to achieve frugal open innovation will reengineer their innovation and performance strategies to meet the economic needs and societal preferences in building a holistic education system towards leaping into Industrial Revolution 4.0 (IR4.0). Therefore, PHEI with right structures and processes will accelerate the ranking of Malaysian private PHEI at the global playfield. Overall, a holistic education ecosystem is the end result of frugal innovation through IT capability and intellectual capability within the context of private PHEI. However, without information technology capability and intellectual capital, frugal innovation will not happen, and without frugal innovation, the transformation of PHEI responding to market shifts induced by IR 4.0 will not happen.

### 2.2. Intellectual Capital

Intellectual capital (IC) plays an important role in PHEI because institutions emphasise non-physical activities where professors are source of knowledge and transfer it to students [27–29]. Knowledge are created through teaching and learning as well as through scientific and technical research. According to [29], IC is articulated as all information, knowledge and experience and intellectual property that belong to a firm. In a recent study, Ref. [3] described IC indicators as follows:

- Human capital (HC): intangible assets that belong to individual capabilities, which consist of the 'expertise, knowledge and experiences of researchers, professors, technical and administrative staff and students' competencies'.
- Structural capital (SC): resources established in the organisation itself, which covers the 'databases, the research projects, research infrastructure, the research and education processes and routines, the university culture, image and reputation' and so on.
- Relational capital (RC): intangible resources' capability to generate value through relationship with the university's internal and external stakeholders. This includes partnership with NGOs, professional bodies, employers, business associations, collaborations with international research centres, interactions with other universities' professors, student exchange, international recognition of the universities, attractiveness, and so on. RC is also known as an important element of social capital. In PHEI, RC includes trust, commitment, communication and association with other HEIs, private as well as public enterprises and the reputation that the academic institution brings within and outside university [30,31]. Basically, RC comprises of total resources that manages all the internal and external relationships of organisation with its environment. In current business environment, it is understood that customers are the most crucial elements for the firm's activities. Thus, relational capital includes business association, communication and relationship with clients [29]. Business are expected to manage their human and organisational capital in order to improve the quality of services to satisfy customers. It is important to note that SC and HC focuses more on internal elements within the organisation whereas relational capital deals with external people and business outside the organisation which has direct and indirect link with the organisation [32].

Universities and industry collaboration and partnership would benefit both "academics and business practitioners" to get mutual benefit. Knowledge is created through teaching and learning as well as through scientific and technical research. Higher expectation to place more emphasis on transferring the intellectual property developed through campus research into the marketplace to stimulate local economic activity and attracting the resources necessary to conduct graduate education and research at world-class levels. Given the autonomy and structural adjustments, PHEI could pay more attention to develop fund raising strategies from private corporates [33,34]. According to [35], a few HEIs in China have adopted innovative approaches to generate higher funds through commercialization of industry-based researches, consultancy projects, outsourcing ICT services, business start-up and so on. Ref. [36] pointed out that HEI not only able to reduce

the reliance on the government, they could also improve their research and development, copyright, patenting and consultancy.

### 2.3. Information Technology Capabilities (ITC)

In rapidly changing environments, accumulating valuable assets through IC is often insufficient to support remarkable performance [37]. Greater performance depends on a firm's capability to 'integrate, build and reconfigure' such intangible resources [38]. The firm's ability to effectively apply its knowledge resources to develop dynamic capabilities that can enable it to quickly response to the environmental changes [39]. Ref. [40] identified three important dimensions of dynamic capabilities. The first dimension focuses on a firm's capacity to sense and form opportunities and threats. The second refers to a firm's capacity to successfully seize identified opportunities. The third captures activities that maintain competitive advantage through merging, defending and reconfiguring intangible and tangible assets. ITC influences how firms acquire competitive advantage and technology information as well as how they translate information and resources into learning activities and strategic actions [41]. Undoubtedly, IT supports businesses, whether through better decision-making ability, growing relationship with customers, business partners and trade suppliers, enabling the computerization of manual activities or advancing organisational innovation. Managing knowledge and IC embedded in the technologies and developing ITC can create value for the organisation. Moreover, recently, the world is witnessing the phenomenon of business digital transformation, which opens up to a new business management, connectivity with external parties, ability to compete globally and the way businesses capture the value. The prevailing knowledge created from IC and ITC, which can create opportunity to be sensed and transformed to improve organizational efficiency and effectiveness.

The literature on management and information systems present comprehensive discussions on how IT can support and improve businesses. Ref. [42] introduced the concept of ITC, which means an organisation's ability to mobilise, integrate and organise IT-based resources. The concept is further defined by [43] as the ability of an organisation to assemble and deploy IT-based resources together with other physical and non-physical resources and capabilities. IT-based resources are IT-enabled resources which comprises non-tangible IT-enabled resources and technical and managerial IT skills including knowledge, assets, customer relationship and synergy across the whole organisation that enable to create a platform for resource, knowledge and capability sharing. Therefore, organisations need to integrate various resources to enhance capabilities that help them enhance their performance [44]. According to [45], organisations will lose to their competitors unless they go through digital transformation by investing in digital technology that has major impact on business improvements such as greater customer engagement and satisfaction, restructuring efficient and effective operations and developing new business models.

Many prior studies have utilised IT expenditure as a measure of IT value [19,46] However, in this research, the focus is on the intangible use of IT capabilities to analyse its influence on innovation and performance. Moreover, [18] examined the relevant uses of information and communication technologies (ICT) with social capital and other knowledge resources able to enhance firm's performance in a knowledge-intensive business unit. As a result, in recent years, many PHEI are developing their own digital strategies by investing in advanced technology, but unfortunately, they do not have sufficient IT literacy [47]. There is also lack of IT capability, commitment or goal to implement digital transformation effectively [48]. Many organisations including PHEI have been facing challenges such as lack of technology integration, poor adoption of technology for strategic use, poor technology understanding and implementation [49–53]. Thus, there is a criticism that most of the institutions invest huge amount in IT systems development that does not able to bring the desired results and value to the institutions [54,55]. Therefore, knowledge, skills and abilities play an important role in creating IT capability, which embedded within an institution and their human capital. Hence, the prevailing knowledge created from

intellectual capital and IT capabilities able to improve organizational capabilities through frugal open innovation.

### 2.4. Framing Direction of ITC Using Dynamic Capabilities Theory

In order to develop and test the conceptual framework, this research draws on knowledge-based view (KBV) and dynamic capability theories to attain and create new capabilities. From the RBV theoretical perspective, exploitation and development of organisations' innovation capabilities should be developed by building internal capabilities on resource utilisation and core competencies rather than simply relying on external market forces and market opportunities for improvement. With its origins in the RBV of the firm, the dynamic capability framework resolves to explain how firms can continuously evolve through responsive modifications made to their resource bases. KBV is an extension of the RBV of the firm [56,57]. KBV emphasizes the increasing role of intangible assets in an organisation, whereas dynamic capability theory focuses on creating competitive advantage in a dynamic modern business environment. Hence, to improve a firm's performance level, organisational learning can be enhanced through attaining and managing knowledge-based resources by accumulating and processing it. ITC is considered as dynamic capability, and it is described as an organisation's ability to create new products and processes in a rapidly changing environment [58]. Moreover, ITC supports innovation processes to create higher productivity rate, improve customer relationship and decrease operational costs. Given the paucity of research on this issue, identifying opportunities to enhance the existing knowledge on IC management in PHEI especially in Malaysia is the key goal of this research. The goal of this research is to fill the gap in the existing literature on PHEI's performance measurement initiatives, underlining the growing importance of IC, which is often overlooked in the academic discourse and by policy makers and marketers.

Using KBV and dynamic capability theory as theoretical lens, this research examines how PHEI can focus on different internal and external resources in IC and IT capabilities, consequently facilitating FOI in a rapidly changing environment (refer to Figure 1).

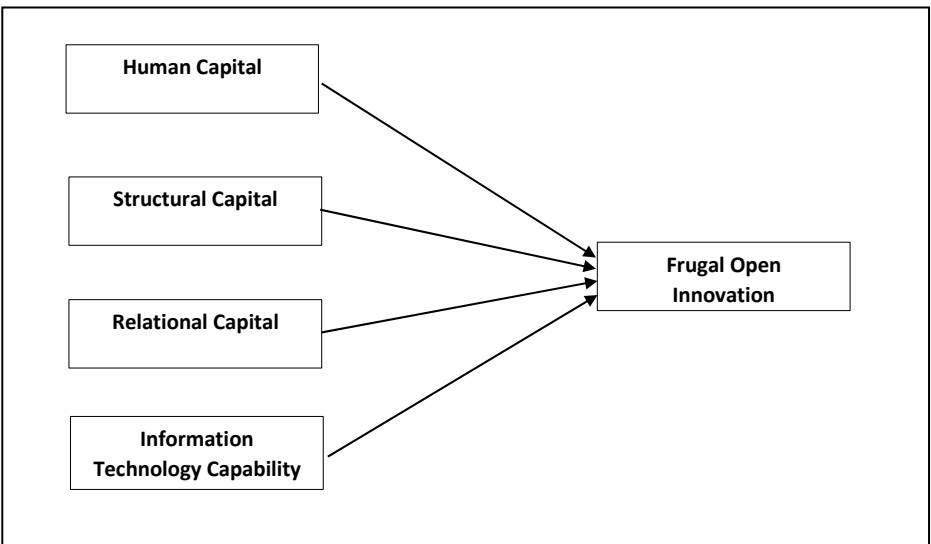

**Figure 1.** Theoretical framework developed for this study.

In summary, this research attempts to answer questions pertaining to (1) the challenges faced by PHEI in current environment, (2) how PHEI develop resources or capabilities to sustain their performance, (3) the need for changes in universities business models or operations in PHEI to be successful in current environment, and (4) how frugal open innovation can play a role in PHEI to face the challenges.

### 3. Methods

This paper presents a practical review aimed to identify the challenges of PHEI from practitioners' perspectives. The paper sits in an interpretive research paradigm. A qualitative approach based on interviews was used with four experienced practitioners about their experience managing PHEI. The four experts selected have extensive experience in managing HEI's administrational task, leadership, and are involved in implementing curriculum transformation, and participated in R&D decision making and contributed to policy making. Discussion with the practitioners in the field offers a chance to identify practical and 'real-world' problems, which can be understudied or ignored within academic research. This approach can provide practical knowledge that gives insight into new directions to make research findings relevant to practice and help in designing future research. Moreover, this approach gives an opportunity for researchers to engage with practitioners and enrich understanding of the challenges faced by PHEI in the current scenario. Four interviews were conducted using semi-structured interview questions, centred on challenges in PHEI with the aim of exploring specific themes in greater depth. Each interview lasted approximately one hour. The interviewees were in managerial position, and their experiences and views added valuable input.

This study combines literature review analysis and explorative expert interviews [59,60] in order to develop the problem statement. According to [59], the sample should involve experts who can be distinguished from 'laypersons', because they have a specific knowledge that enables them to provide good insights for the findings. Thus, interviewees with various institutional backgrounds in PHEI, as well as interested and active people, such as an academic consultant, top management and university level management, were included in the paper.

Our purpose is to develop clear problem statement that was evident from extensive literature review about a specific topic but because of an aim of developing a socially meaningful and researchable problem statement that is original, and practice review through expert interviews with four experts (refer to Table 1). This type of expert interview is uniquely aimed at obtaining reliable data because respondents' competence is very high [61]. Through discussions with the experts in the field, it will provide insight to identify real world issues and make the research findings relevant to practice. On the other hand, ref. [62] mentioned that qualitative researchers always work with small size sample of people within the context and studied in depth contrary to quantitative studies who aims to generalize to a larger sample size and seek statistical significance.

Three styles of expert interviews are defined by [59]. The exploratory expert interview is the first kind, and it is commonly used to acquire expertise and orientation in new or unfamiliar fields. This aids in the organisation of a diverse field and the generation of initial hypotheses. The systematising expert interview (also known as the exploratory expert interview) is the second kind. This form of expert interview seeks to gather expert information in a standardised and systematic manner in order to achieve a high degree of data comparability [63]. The last one is theory-generating expert interview, as described by [59], serves as a starting point for the planned methodological creation of problem-centred expert interviews. The problem-centred interview (PCI), which was introduced by [64], is a qualitative face-to-face interviewing technique that incorporates key qualitative research concepts like transparency, versatility, and process orientation [65]. This paper uses problem-centred expert interviews to get deeper understanding on the current situation. Meanwhile, further studies will be conducted academicians and management in PHEI to obtain more contextual knowledge.

**Table 1.** Comparison of critical literature review and practical review.

| Area | Influence |
|---|---|
| 1. Critical appraisal of literatures | The critical appraisal of previous studies that appears in journal articles may suggest a problem area for the researchers to focus and stimulate thinking. The researchers only able to focus data from a specific group or country that can provide a basis or guidance for developing a conceptual framework to determine the strength of relationships in their own research context. A research idea also maybe suggested through research gap identified in past literatures and future study areas suggested. |
| 2. Practical experience review | Practical experience shared by the experts will provide wealth of experience from which a research problem can be derived and strengthened. The researchers may observe a particular incidents or events occurring and curious why it happens and how it contributed to current state. Brainstorming with experts will provide a chain of thoughts and valuable feedback to the researchers to focus on a specific research questions and explore further the specific problem. |

The Figure 2 below illustrates the mapping process of literature review and practice review to derive problem statements and the findings towards insights, challenges, conceptual model and research gap.

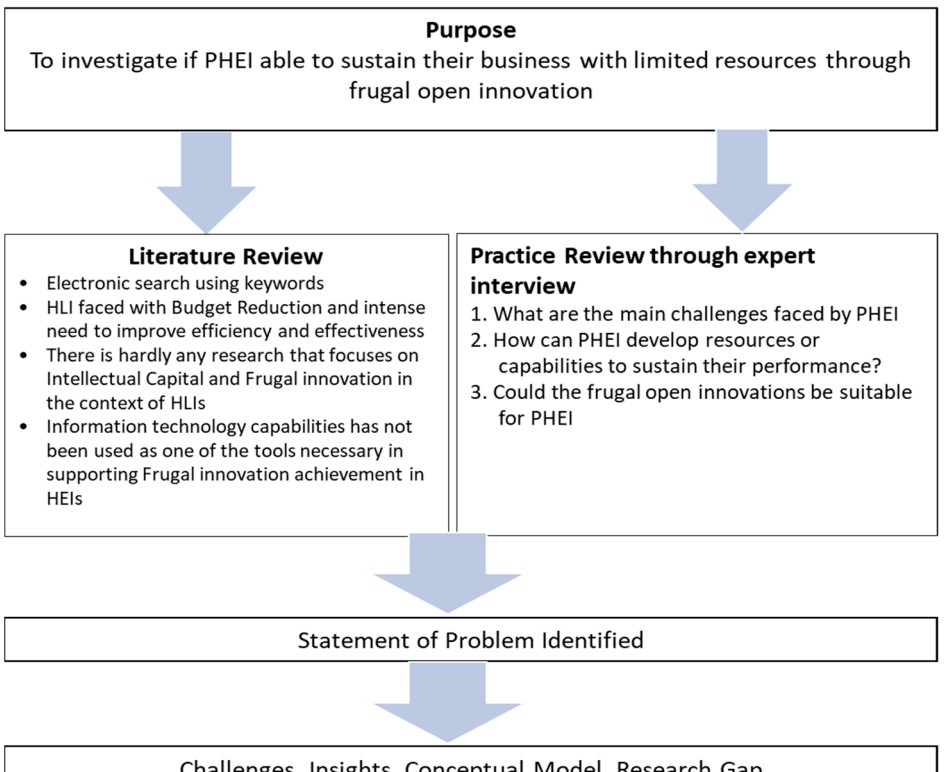

**Figure 2.** Mapping the literature review and practice review to develop problem statements and findings.

The scope of this study is limited to PHEI although Malaysian higher education institution is represented by private and public HEI. Public HEIs consist of public universities, polytechnics and community colleges while the PHEI, inclusive of private universities, private university-colleges, foreign university branch campuses and private colleges. There is a huge variation amongst HEI whereby there are private institutions with substantial private funding and public institutions, which rely heavily on government support. Apart from that, among the PHEI, colleges are operated in smaller scale, have different missions and visions and offers lesser range of courses as compared to universities. PHEI have been awarded the university status by Ministry of Education as it has fulfilled the criteria of substantial amount of research activities, conducive learning environment, financial viability and offer various undergraduate and postgraduate courses. Therefore, PHEI have to utilize their potential for aggressive resource seeking, cost efficiency and generating profit for survival due to intense competition and transformation. Research on PHEI is in line with the Government's initiatives to strengthen the competitiveness private education and improve quality of the PHEI.

Purposive sampling technique was used. Key informants were chosen based on their expertise and affiliations, recent publication in the field of PHEI and knowledge about the issue (refer to Table 2). The practitioners were contacted and invited to participate through e-mail. The variation in the positions of the interviewees provides different perspectives. The interviews were recorded and transcribed verbatim right after the interview session. The semi-structured interview allowed the practitioners to present subjects that were more important in their view that led to subsequent questions. Finally, broader aspects were covered. The practitioners preferred to stay anonymous, so the names of the practitioners and PHEI were not disclosed when quoting their responses. The initial data collected on the challenges were then grouped into five challenge themes and presented. The findings were also presented in the form of discussions based on the practitioners' explanations. In terms of credibility, practitioners were selected on the basis of their vast knowledge and experience on the subject matter. Practitioners were comfortable sharing their views in their regular environments, which increased the internal validity of the research. The paper highlights the unique value and contribution that this approach offers by conducting a practical review to enhance the understanding of the literature review.

**Table 2.** List of Practitioners.

| Expert | Position | Credibility |
|---|---|---|
| Practitioner 1 | Quality Assurance Consultant, Former Deputy Vice Chancellor of HEI, Former Deputy CEO of Quality Assurance Agency | 30 years of experience<br>Role: COPPA Facilitator<br>Quality Assurance Programme and Institutional Assessor |
| Practitioner 2 | Vice President<br>PHEI 1 | 23 years of experience<br>Role: Research, Industrial Collaborations and Engagement |
| Practitioner 3 | Associate Professor, Dean,<br>PHEI 2 | 25 Years of experience<br>Role: Faculty Administrative Leader, promoting academic, research and innovation, engaging external collaboration, and accreditation |
| Practitioner 4 | Head of Section, Foundation Business,<br>PHEI 3 | 20 years of experience<br>Role: Academic leadership,<br>lead, manage and develop department |

Interview transcripts were analysed using NVivo, a qualitative analytical software tool. We used word frequency analysis and obtained word clouds with list of terms to form themes from the responses of all four experts for each questions.

### 4. Results

The specific objectives of the research are as follows: (1) to identify the current and future challenges faced by private universities, (2) to know the current changes in PHEI and (3) to obtain practitioners' opinions on their perceptions towards PHEI's business model. The challenges highlighted by the practitioners were grouped into five main categories, which are structural challenges, operational challenges, financial challenges, social challenges and technological challenges. Tables 3–5 show the five challenges faced by private universities. All challenges compiled from the interview are discussed in detail in the following subsections.

**Table 3.** Structural Challenges in PHEI.

| Practitioners | Interview | Key Findings |
|---|---|---|
| Practitioner 1 | "Some of small colleges do not have sufficient infrastructure and facilities to cater diverse need of their users. Even they have, it is costly to maintain building and develop facilities. Not only that there is a need to manage energy use, waste disposal, resource use and environmental pollution/contamination" | • insufficient infrastructure<br>• costly to maintain building and develop facilities<br>• managing environmental, energy and resources |
| Practitioner 2 | "I feel university is undercapitalized. In order to support modernization, we need to invest in long-term facilities either through additional income generation or through borrowing. Another issue that I want to share is management's view that this facility is be treated as part of the operational costs and must be minimized, rather than a strategic asset that must be optimized to eliminate redundancy." | • undercapitalized<br>• invest in long-term facilities either through additional income or borrowing |
| Practitioner 3 | "I think universities have spent a lot of cost to prepare maintenance in terms of facilities and building. It is very costly. But what management feel is that eventually, it is about managing the facilities with less cost and creating more value. I mean more ROI. There is always limited budget for capital expenditure as difficult to convince using cost benefit analysis" | • spend substantial parts of their expenditure on building maintenance<br>• facilities need to be managed with minimum cost rather than optimum value<br>• ceiling or limitation on capital expenditures |
| Practitioner 4 | "See, most of the time capital expenditures by faculty or departments are compared against their contribution to the universities' income. That means the faculty that contribute higher to universities' profit will get capital investment approval easily".<br>"I also notice problems like connection problem, not enough equipment and facilities are outdated".<br>"Universities always have dilemmas like whether they should keep and maintain old building or they should upgrade or replace the facilities because it is very hard as they need to create a safe workplace and they have other uses for the facilities labs, library etc, they should eliminate redundancy in asset use." | • Lack of infrastructure/equipment, connectivity.<br>• Insufficient facilities dilemma in managing aging buildings and infrastructure—"retain and maintain" versus "upgrade or replace"<br>• catering for the diverse needs of users of the facilities and infrastructure<br>• optimising the utilisation of space, plant, equipment and grounds; elimination of redundancy in asset use |

**Table 4.** Operational Challenges in PHEI.

| Practitioners | Interview | Key Findings |
|---|---|---|
| Practitioner 1 | "Some universities do not have readiness and preparedness. Some of the colleges do not have good lecturer as they pay low, they just get people who wanted to have an income. Not necessarily prepared to teach or become a lecturer. When this hit is a new medium/norm now, they have a whole bunch of people who are very young, not experienced, not motivated, not passionate. Then it begins to show no preparation and readiness. Where there are able to pay better, then could recruit better people, get experienced people and also train them. Because it is a pandemic, most universities trying to make the best out of the disruption." | • lack of human capabilities |
| Practitioner 2 | "Of course, from financial issues it will then translate to some operational issues. Any commercial entity one of the fastest ways to save the boat from sinking is either to have cost cutting or increase revenue. It is almost natural for universities to take up cost cutting as their first measure in providing financial stability. As a consequence of cost cutting, operational issues are affected. Budget for different departments and faculty will be reducing like operation not operating same as before, ranging from services and actual resources that we bought becomes lesser. There will be consequences to services, staff morale and efficiency due to cost cutting." | • cost cutting measures affects<br>• operational issues<br>• attracting and retaining the best talent<br>• budget to departments and faculty will be reducing |
| Practitioner 3 | "It is really challenging to manage our core business, maintain efficiency and align the institutions strategy and visions together in current scenario. PHEI success depends very much on staff capabilities . . . how we manage staffs' knowledge and skill. So much of dilemma in deciding how much budget should be allocated for various functions such as manpower, facilities, faculty activities, software, and hardware while satisfying various and conflicting university goals." | • managing staff capabilities.<br>• difficulties to allocate budget to various department |
| Practitioner 4 | "The moment universities face insufficient funding, overstaffing in non-academic areas and lack of monitoring and control mechanisms, so much of operational issues might arise". The have issues like poor capital management and inefficient budgetary allocations for operational activities...usually the management have so much of things to settle with tight budget and less resources like skill shortage". | • poor capital and operational<br>• budgetary allocations<br>• facilities are operational costs that must be minimised, rather than a strategic asset that must be optimised<br>• skill shortage and the need to do so much with a tight budget. |

**Table 5.** Financial Challenges in PHEI.

| Practitioners | Interview | Key Findings |
|---|---|---|
| Practitioner 1 | "The biggest issues for the university are survival planning not even doing strategic planning. How to survive because the number of foreign as well as local students going down. For those universities rely significantly on foreign student is completely disaster". "The government did not extend too much support to this sector. The university without sufficient financial buffer are hard hit, they are letting go their staffs, cutting down their staffs, letting go some of the overhead which are not considered as non-essential stuffs." | • cash flow challenges<br>• sharp fall in enrolments over the past few years<br>• inadequate public funding<br>• price sensitive consumer<br>• increasing operational expenses |
| Practitioner 2 | "At the moment, the biggest challenge for universities is financial sustainability. As a private university of course, we are providing our core business which is education to school leavers and prepare them for the workforce." | • cost is increasing rapidly<br>• facing difficulties to bend the cost curve and increase the productivity of the education<br>• payroll costs are a big challenges<br>• improve business operations against performance objectives<br>• increasing revenue in short run might increase more cost |
| Practitioner 3 | "Private universities facing constraints and difficulties because they are very much depending on students' fees. A lot of issues faced recently like how to fund higher education, managing rising costs and competing for students, talented staff and grants. University needs to shift the focus on other method of generating income and not depending on students' fees as student enrolment has dropped generally which gives pressure in terms of financial to the universities. So, focus more on controlling expenses and allocating budget resources effectively and efficiently is the biggest challenge. Sometimes we need to spend less on non-essential activities" | • how to fund higher education<br>• rising costs and reduced funding |
| Practitioner 4 | "I believe that most of the universities are facing cost of a traditional education continues to increase, enrolment declines, reduced revenue streams and lower operating margins.in other word less money but being chased by many PHEI. Not only that, I think universities struggle to stay in business under their current model. So, their biggest challenge is to lower cost without sacrificing their corporate value to address the stakeholder needs. Maybe they need to develop simpler and affordable programs and services. Every university has their own mission. So basically, to create best educational and research programme and make some profit, university must set their mission and vision and align their strategy towards achieving those goals and priorities." | • cost of a traditional education continues to increase.<br>• enrolment declines, reduced revenue streams and lower operating margins.<br>• universities struggle to stay in business under their current model.<br>• optimizing the utilization of space, plant, equipment and grounds; elimination of redundancy in asset use |

### 4.1. Structural Challenges

Most of the facilities and infrastructure in PHEI are either invested through income generated or borrowings. Practitioner 3 claimed that most of the time, senior management would be more interested to know how much of a return they would be receiving on facility investment rather than how much money is being saved. She argued that when it comes to requesting for capital investment budget, the senior management will no longer be convinced by justifying cost savings. They will be more interested to know the business value that will be able to contribute in terms of financial return. Practitioner 4 added that capital expenditures made by cost centres or departments are traced directly to the department in proportion to their contribution to universities' income generation. Practitioner 4 also shared that there is always a big dilemma in managing aging building and infrastructure, that is, whether to 'retain and maintain' or 'upgrade or replace'. Therefore, the departments or activities that make significant contribution to universities' competitiveness and profit will stand better chances getting capital development approval.

Practitioner 1 commented that universities especially private universities have insufficient infrastructure and facilities to cater the diverse needs of their users. He also highlighted that most of the universities spend substantial amount of their income on building maintenance expenditure. Apart from building management, according to Practitioner 1, other issues must be given importance by PHEI such as energy use management, water waste management, resource utilisation and environmental protection. Practitioner 2 argued that most of the time, the management is always concerned about managing facilities at minimal cost rather than optimising value through maximum utilisation and elimination of redundancy in asset use.

The word cloud generated using NVivo software (refer to Figure 3) also indicates that facilities, building, capital, income and cost are among the most prominent word being mentioned by the practitioners. The finding conforms with the issues highlighted earlier that facilities and infrastructure can be costly and requires wise budgeting.

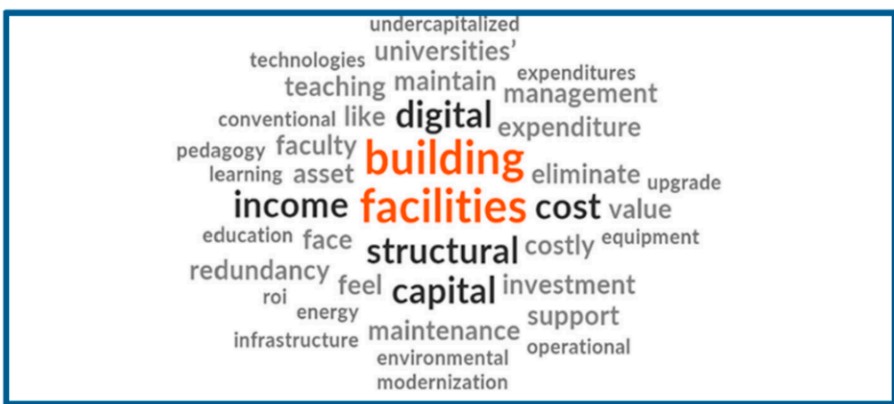

**Figure 3.** Word cloud diagram for structural challenges.

### 4.2. Operational Challenges

According to Practitioner 2, managing cost and increasing revenue are two big challenges in private PHEI that highly rely on fees paid by students as the number of enrolments declines. Practitioner 2 also felt that many PHEI are struggling to sustain their business and continuously provide quality educational service as the cost cutting measure cuts down most of the operational budget, leading to consequences to services, staff morale and efficiency. Practitioner 3 highlighted 'it is really challenging to manage our core business, maintain efficiency and align the institutions strategy and visions together in current scenario'. She claimed that PHEI's success depends ultimately on staff capabilities and the ability to manage staff's knowledge and skills.

Practitioner 3 explained that there is much dilemma in deciding how much budget should be allocated for various functions such as manpower, facilities, faculty activities,

software, and hardware while satisfying various university's multiple and conflicting goals. Practitioner 4 highlighted that the major problems are insufficient funding, overstaffing in non-academic areas and lack of monitoring and control mechanisms. Practitioner 1 stressed that effective resource allocation is required to solve the continuing difficulties faced by universities. Hence, he added that an organised allocation of available resources is needed to best allocate resources among various functional areas.

In order to extract the most common themes in operational challenges, word cloud was generated and analysed (refer to Figure 4). The most frequently used words are operational, people, teaching, learning, services and students, which reflect that the major concerns in operational issues are teaching and learning related. Surprisingly, given HEIs' main role in teaching and learning activities, experience and skills were less mentioned and appeared as minor terms.

**Figure 4.** Word cloud diagram for operational challenges.

### 4.3. Financial Challenges

All the practitioners claimed that the biggest challenge to PHEI is increasing expenditure and shrinking funding. Practitioner 3 mentioned that private universities face resource constraints as they very much depend on students' fees. Unfortunately, student enrolment is decreasing lately which causes financial pressure to the universities. Therefore, universities prioritise controlling expenses and allocating resources effectively in their budget. According to Practitioner 1, PHEI should develop programmes and services, which are affordable, fulfil the need of job market and have lower cost. He added that the way forward is to use technology-based teaching and learning methods.

According to Practitioner 2, PHEI should use lower margin and higher volume business models. He mentioned that 'we always compete for more students, faculty acquisition, staff retention, funding and achievement of research or teaching goals'. Practitioner 2 commented that universities are struggling to best allocate their resources and use these resources effectively. There is always a dilemma on how to educate more students with relatively lesser resources and lesser cost to deliver quality education. Practitioner 4 claimed that the biggest challenge is to lower cost without sacrificing the corporate value to address the stakeholder needs with simpler and affordable programmes and services. When asked to explain further on financial constraints, Practitioner 1 explained that it is important to keep the university's programmes and services simpler and affordable.

The findings from the word cloud analysis (see Figure 5) conform to our findings that cost, revenue and financial elements are the top mentioned terms. Affordable, quality, fees and performance are also mentioned quite frequently.

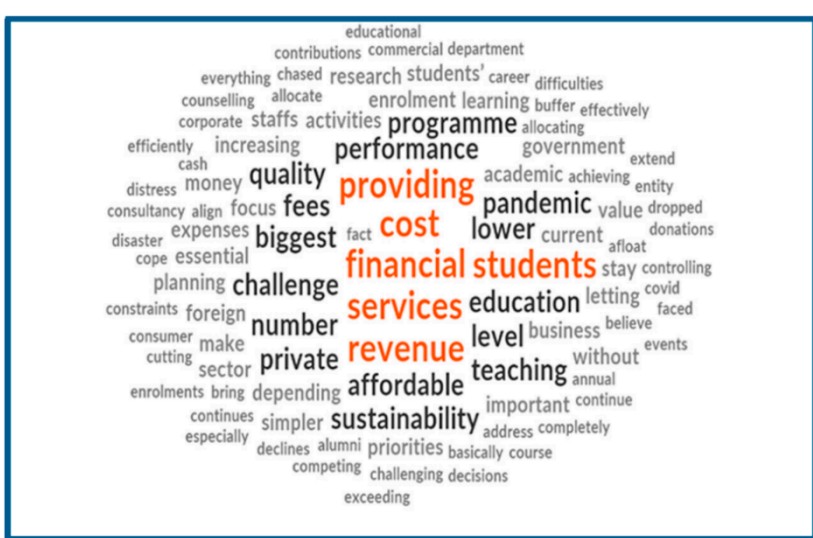

**Figure 5.** Word cloud diagram for financial challenges.

### 4.4. Social Challenges

Based on Table 6, practitioner 4 claimed that consumers in PHEI are generally more value conscious with various needs and costs. The biggest challenge faced by PHEI currently is lack of academic staff capability, which affects the innovation capability among universities. Practitioner 2 added that recruiting as well as retaining talented staff is another big challenge for PHEI. Many PHEI implemented organisation-wide freeze on employment due to insufficient budget. Practitioner 1 added that most of the time, PHEI are understaffed and do not have the right talent which makes them operate at only one third level of their strength.

One of the biggest challenges faced by private universities according to practitioner 3 is resource scarcity, in particular attracting talented manpower especially those with innovation and research skill. PHEI continue to face challenges in providing conducive and safe working environment that can support greater efficiency in the vital activities of the institutions. Practitioner 1 commented that due to the changing market conditions, universities are being criticized for being unable to provide more flexible, seamless and personalised educational experiences. Owing to the lack of corporate image, private universities cannot sustain competitive advantage to attract world-class academics and top students. Moreover, according to Practitioner 4, weak intellectual property (IP) protection and knowledge sharing with various partners also present extra challenges to universities. She also commented that there is no significant efforts towards promoting sustainability at university operations.

The larger terms in the Figure 6 shows are those mentioned frequently. Words which are frequently mentioned include image, academic, capability, experience, skill and employment. These terms are related to human capabilities. Issues on customer expectations, collaborations and community engagement are also highlighted as part of social challenges.

**Table 6.** Social Challenges in PHEI.

| Practitioners | Interview | Findings |
|---|---|---|
| Practitioner 1 | "Universities faces value-oriented customer with various needs. Many universities will face organisation-wide freeze on employment and inadequate budgets, nowadays it has been very difficult attracting and retaining skilled manpower. Universities are understaffed and do not have the right talent which makes them to be able to operate at only one third level of their strength." "Due to the changing market conditions, universities are being criticized for not be able to provide more flexible, seamless and personalized educational experiences. Due to lack of corporate image, private universities could not sustain competitive advantage to attract world-class academics and top students." | • value oriented customer with various consumer needs<br>• organization-wide freeze on employment and inadequate budgets<br>• attracting and retaining skilled manpower |
| Practitioner 2 | "We know that there is always rising student expectation but reduced number of staffs. Recruiting as well as retaining talented staffs is also one of the very big challenge for PHEI. Many private universities implemented organization-wide freeze on employment due to insufficient budget". "There is a problem like increasing student mobility because now student able to study anywhere they want due to expansion of higher education marketplace." | • rising student expectation<br>• rising number of students but reduced number of staffs |
| Practitioner 3 | "Attracting staffs, students and industry collaboration is very challenging. Another big challenge is having fewer academic staff especially those who are equipped with research skill and innovative skill. Another challenge is to provide safe and healthy work place that can improve staff's efficiency in the university. University must keep close relationship with community, industry, professional bodies and government. We must have good deal of work or research output impact to maintain the partnership". | • academic staff capacity<br>• challenge of providing conducive, safe and healthy work environment that supports productivity and excellence in the key activities of the institution |
| Practitioner 4 | "For social aspects, like I said before, resources are scarce. I think students nowadays looking for more flexible, continuous and personalized educational experiences that can most fit to their preferences. Of course, as we know there is there will be continuing influence of changing economics and market conditions in current education landscape. Consumers in PHEI are more value conscious with various need and cost. Issues like lack of academic staff capability will affect the innovation capability among universities." "PHEI also have lack of corporate image and competitive edge to attract top students and world-class academic". I also feel weak IP protection and knowledge sharing with various partners also gives extra challenges to the universities. It looks like there is no significant efforts towards promoting sustainability at university operations." | • resource scarcity<br>• students looking for more flexible and personalized educational experiences<br>• continuing influence of changing economics and market conditions<br>• lack of corporate image and competitive edge to attract top students and world-class academics |

**Figure 6.** Word cloud diagram for social challenges.

*4.5. Technological Challenges*

PHEI face a few issues in terms of technological challenges as highlighted in Table 7. Practitioner 1 highlighted that private universities lack not only IT infrastructure but also technological capabilities. He added that the current biggest challenge for private universities is to keep their staff abreast of digital platforms to innovate learning and teaching experience and current technology advances. He added that there is a need bring research to next level by using big data and artificial intelligence. Unfortunately, according to Practitioner 1, only teaching and learning, library and some administration jobs have adopted digitalization, but the other parts of universities remain the same. The key force in their education model is still using residential, in-campus and classroom as their value proposition despite investments to digital resources to be adopted in teaching. Practitioner 2 shared the same opinion that universities face technological obsolescence and equipment and digital platforms must be upgraded. Practitioner 3 also stressed that universities need to keep updated with current evolving technologies. According to Practitioner 4, 'technology is a root for innovation. However, unfortunately, many universities do not have skilled staff and lack good technological capabilities which become roadblocks for innovation capability'.

The word cloud provided in Figure 7 indicates that changing skill, experiences, flexible capabilities, employment, challenges, personalised and market needs are important concepts in technology capability. These terms are very consistent as PHEI are surrounded and targeted by ever-changing digital and automated services that are embedded in their daily operation but on a patchy basis. For PHEI to grow their mission and impact, they have to focus on the digital domain.

**Table 7.** Technological Challenges in PHEI.

| Practitioners | Interview | Key Findings |
|---|---|---|
| Practitioner 1 | "Some of the small colleges has lack of IT infrastructure. Not only that, most important issue is not having sufficient capabilities in technologies. We can see universities really working hard and facing challenges to keep staff abreast on current technology advances to innovate teaching and learning experience, changes in environment and legislations. Anyway, there is a need to develop and undertake research to next level by using big data and artificial intelligence." | • lack of IT infrastructure<br>• insufficient capabilities in technologies<br>• keeping staff abreast of current technology advances and changes in legislations |
| Practitioner 2 | "Yes. IT capability is very important to increase innovation capability" I think the universities facing technological obsolescence and there is a need to upgrade equipment and digital platforms. I think that it is time for PHEI to rethink their education model and tie it together with the technology to deliver cost effective education and revolutionize research through the use of big data and artificial intelligence. He said, "most importantly, ICT could help universities to optimize customer experience, grow their revenue and lower the cost." | • need digital platforms to innovate the teaching and learning experience<br>• revolutionize research through the use of big data and artificial intelligence<br>• technological obsolescence and the need to upgrade equipment and processes<br>• IT capability is very important to increase innovation capability |
| Practitioner 3 | "We cannot use outdated technology platforms and must keep up with the changes introduced by the evolving technologies. It is very drastic. We have no choice especially due to this pandemic, our staffs are working from home, using remote teaching and we don't meet physically. We started to use online platform for teaching and learning, meetings, research collaboration and assessment. Surely there are a lot of hick ups at the beginning. Of course, we need time to continually improve the efficiency of new ICT developments, but training and support are given to the staff." | • keeping up with the rapid changes introduced by the evolving technologies<br>• outdated technology platforms |
| Practitioner 4 | "In my opinion, traditional universities cannot sustain and need to look at new business model with online or hybrid learning especially now during pandemic." "The issue now is how to leverage new and efficient technologies to improve on the management and operations." "Even though technology is root for innovation. But unfortunately, many universities do not have enough budget, skilled staff and lack of good technological capabilities that becomes obstruction for innovation capability." | • conventional universities unable to sustain and need to look at new business model with online and hybrid learning<br>• how to leverage new and efficient technologies to improve on the management and operations |

**Figure 7.** Word cloud diagram for technological challenges.

## 5. Recommendations and Suggestions for Changes of Business Models or Operations in PHEI to Be Successful in Current Environment

Based on Table 8, all the practitioners believed that the COVID-19 pandemic has speeded up the digitalisation of PHEI. Practitioner 1 mentioned that the pandemic has exposed how badly and unevenly PHEI was warming up to this shift. According to him, the PHEI are probably the only institution without any major disruptions for a long time, and their education model is still mainly focused on in-campus, whiteboard, scheduled classes, rostrums and classroom. Some elements of digital resources are introduced to support teaching, but building and facilities/real estate is the core aspect of the value proposition. There is a strong belief and preference on conventional campus-based big brands, in-person pedagogy or face-to-face teaching as it has social, cultural, personal and religious factors. Online delivery is only provided by not branded or prestigious universities and is rather only used by distance learning or open universities in Malaysia. According to him, 'a new education strategy which is aligned with a new digital vision and mission which requires more investment in digital technologies are needed'.

**Table 8.** Recommendations for PHEI.

| Practitioners | Interview | Recommendation(s) |
|---|---|---|
| Practitioner 1 | "Universities have to develop their strategies, redefine their operating models and to implement major technology transformation". "Universities and colleges are probably the only institution without any major disruptions for long time and their education model is still mainly focused on in-campus, whiteboard, scheduled classes, rostrums and classroom. Universities need to build new capabilities as the education has changed a lot even there is no pandemic." | • redefine strategies, operating models and implement major technology transformation. |
| Practitioner 2 | "I think private universities especially in Malaysia definitely has to change in business model altogether. And depending on student revenue from student fees is almost not sustainable anymore . . . not in Malaysia. I think one of the business models that the university can use is endowment or wakaf. I think this is one of the methods that MOHE is advocating. To survive this call and to face today's challenges, it requires innovation all the way down to the business model and not only focusing at operational or strategy level because it won't be enough if we only change the strategy. Another one of the trait or characteristics of the new business model is to focus at using less resources and doing more but it is not sufficient where there should be also other way like endowment . . . definitely we will look at cost cutting and next natural step is to increase the revenue. But sometimes some of this new revenue will incur further cost." | • focus on the student experience, technology transformation, strategy and policy reform<br>• requires innovation in business model, change only strategy is insufficient<br>• reduce dependence to student's fees and look for auxiliary income<br>• to look into wakaf or the endowment model<br>• achieve 'more using less' model by using existing resources to generate new stream of revenue and existing revenue |
| Practitioner 3 | "Universities need to look at survival and development opportunities as what other private industries are doing. There is a lot of resources shortage but at the same time operational, academic and research cost are increasing, so administrators need to look at managing resources efficiently. Meanwhile, universities quality depending on university ranking, research publications, patent innovation production and grants, affiliation with reputable university and university industry collaboration. Rapid changes are taking place in the business environment now, we need more vibrant business model that looks at prioritizing expenditure, allocating resources more efficiently and exploring new funding for universities". | • business models have to be rethought and introduce new digital models that prioritize allocation of resources, reducing expenditure and exploring new fund. |
| Practitioner 4 | "The universities are facing many challenges to take leading role in digitalization initiatives, even if the university have access to the internet and IT infrastructure for various types of teaching and learning applications but still are reluctant to adopt fully digitalization approach. There is lack of staff's support and involvement, lack of technical knowledge and specialist staff and little information technology training for staff. These issues are related to information technology capability of the PHEI, and academicians become reluctant to adopt digitalization, which is relatively new to them, due to perceived uncertainties that resulted from their past experiences dealing with online platforms." | • technology could be used to reduce the cost of delivery and to grow market reach<br>• develop a strong student and wider stakeholder value proposition |

According to Practitioner 3, private universities are facing many serious pressures arising from the pandemic. According to her, 'of course, all have to continue their sessions online with the flexibilities allowed by ministries and regulators. But we in university still desperately wants our students and lecturers to return back to campus. Otherwise, students might ask for fees reductions and student enrolment will drop tremendously'. She added that this will add to more challenges for these financially distressed institutions.

According to Practitioner 2, the century-old model of PHEI is under criticism. The online version with more options for the learners is gaining higher popularity. She added that it is time for PHEI to rethink their education model and tie it together with the technology to deliver cost-effective education to more. He said, 'most importantly, ICT could help universities to optimize customer experience, grow their revenue and lower the cost'.

Practitioner 4 believes that teaching and learning, internal management and library have adopted some digitalisation, whereas the rest of the university function have remained out of this zone. This ad-hoc adoption or patchwork does not provide sufficient effort to become digitalised service providers. She also claimed that unclear regulations and enforcement on data protection policy concerning information published online and data security can become threats in the future.

The universities are facing many challenges to take leading role in digitalisation initiatives, even if the university have access to the internet and IT infrastructure for various types of teaching and learning applications but still are reluctant to adopt fully digitalization approach. There is lack of staff support and involvement, lack of technical knowledge and specialist staff, and little information technology training for staff. These issues are related to information technology capability of the PHEI, and academicians become reluctant to adopt digitalisation, which is relatively new to them, due to perceived uncertainties that resulted from their past experiences dealing with online platforms. The staff believe that online teaching and learning has some difficulties such as unstable connection, complex and ambiguous processes, lags/delays and limited ability to alter (Practitioner 4).

The word cloud in Figure 8 generated from the interview text shows that some of the important terms are teaching and learning, technology, revenue, business, cost, environment and digitalization. The figure clearly indicates that the new business model for PHEI needs to emphasise teaching and learning excellence, financial sustainability digitalization and technology.

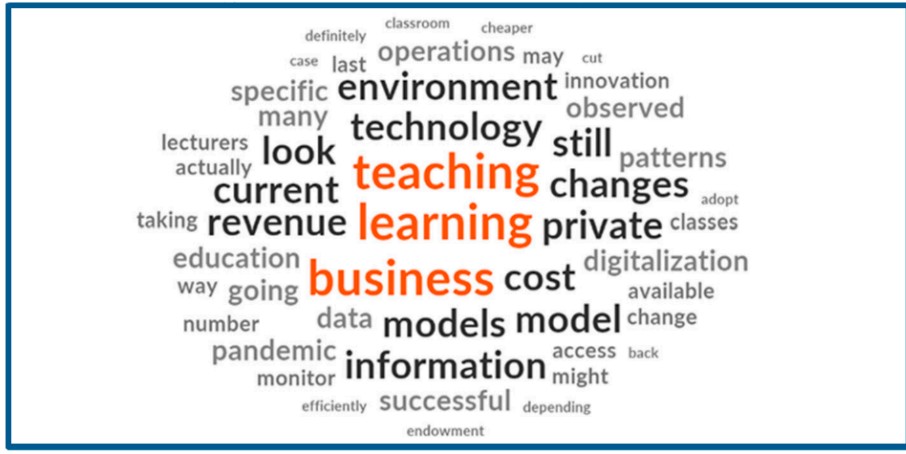

**Figure 8.** Word cloud diagram for PHEI's business model.

### 6. Analysis: Challenges of Transitioning PHEI into Value Creation Using FOI

The above themes can be categorized into four quadrants of challenges that can be seen in Table 9. The preceding section looks at technology as a standalone challenge, it can also be grouped, in our view, alongside operations as technology and the PHEI information systems in general. As such, technological challenges are pervasive and can cut across the value chain.

How can these challenges be tackled? How can systemic thinking and management of PHEI lead to FOI that can address these challenges? Sales and operating planning (S&OP) may be a panacea. Table 8 suggests that for FOI to work, a rethink of the PHEI value chain would be necessary. This includes issues such as rethinking how best talent can be brought in/retained to deliver programs that are in demand. In addition, strategic initiatives such as new product/program development, creative use of limited funding, and deep dive into streamlining business processes might be required. In this regard, viewing PHEI in the lens of sales and operating processes (S&OP) is called for.

**Table 9.** Four quadrants of challenges.

| Challenge 1: Structural | Challenge 2: Operational |
|---|---|
| • Need to ensure sufficient utilization of building capacity.<br>• Innovative use of infra resources to generate alternative income streams.<br>• Sound management and maintenance of facilities given a limited financial allocation for this purpose. | • Need to ensure seamless conversion of products/ideas (curriculum) to sellable programs.<br>• The value chain drivers would include other issues such availability of talent, proper infrastructure, and technical platform to support program delivery of outstanding programmes.<br>• The PHEI value chain would need to have superior front end sales and marketing teams that are capable of luring in customers using innovative platforms. |
| **Challenge 3: Social** | **Challenge 4: Financial** |
| • Core issues in this context entail the need to decipher on how best to manage limited talent with students' demand and expectations.<br>• Practical outcomes would be achieving an optimal balance between infrastructure investments, talent development and a steady growth of student population who feel that the establishment is still relevant. | • Core challenges include high labour cost 'vis-à-vis' declining revenue from core business (student enrolment).<br>• Situation is further exacerbated given limited public sector funding.<br>• These issues lead to practical challenges such as talent retention, reduced compensation, etc. |

### 6.1. S&OP Processes as a Panacea towards FOI

S&OP processes, albeit largely used in the fast-moving consumer goods (FMCG) space, can have significant impact on other industries as well. S&OP processes, when executed well, can lead to FOI and enhance PHEI's overall value and supply chain management. A failure to track and closely monitor technological changes (amid other changes in a volatile, uncertain, complex and ambiguous world) can lead to the demise of even highly successful enterprises. The Global Center for Digital Business Transformation offers a powerful way of examining industries affected by digital disruption. The researchers used the analogy of a vortex to describe industries affected by digital disruption.

A vortex is essentially a force akin to a whirlpool that can suck elements into its core. This definition implies that industries close to the core of the vortex are at the greatest risk of being disrupted. Conversely, these industries also end up leading digital disruptions. The industries closest to the epicentre of the vortex are from the technology sector, such as Facebook, Amazon, Apple and Google. Either these companies provide the relevant tools and technologies for digital disruptions, that is, they enable disruption for others, or they themselves constantly generate new value propositions in the economy and lead the disruption.

The education industry is not spared from digital disruption. Continuous changes in teaching and learning activities coupled with emerging digital platforms that provide self-learning tools and technologies make this sector ripe for disruption. Although in the case of emerging economies such as Malaysia, digital disruption in the education sector is contingent upon strong regulatory conditions, which can prevent an immediate 'sucked into the vortex' syndrome. Nevertheless, academic institutions should constantly track and monitor technological changes that can impact this sector, particularly in ensuring that local graduates remain relevant in the digital economy and IR 4.0 space. S&OP processes, when implemented well, can foster greater agility, an innovative culture and overall business sustainability. In this piece, we examine the relevance of S&OP processes in the context of leading education institutions.

### 6.2. What Is S&OP Processes? How Do They Lead to FOI?

S&OP refers to leading businesses based on an integrated business management process. The term 'integrated' refers to leadership having a holistic supply chain outlook of the business and encompassing core elements such as demand management, sales and marketing, production and new product development. An S&OP leadership view of any organisation is crucial as it enables leaders to achieve (when executed well) the following corporate/business outcomes [26]:

- Demand-driven production—ensuring that products and services developed and offered by the business are in line with market trends and demand.
- Customer-centric solutions—processes are established to ensure a business can pivot itself with agility, meeting exactly what customers want.
- Deeply connected business processes—integrated processes, which include good alignment between product planning, sales, finance and demand management among others.

### 6.3. Applying S&OP Processes towards FOI in PHEI

To ensure long-term sustainability, education institutions must remain relevant. The programmes or courses offered must be in line with industry trends and requirements. In this context, leadership within such institutions can think about six steps inherent in the S&OP processes. These six steps, and how they relate to leadership in an educational context, are summarised as follows, based on Anaplan and Olive Tree Group.

#### 6.3.1. Product Review

Essentially, product review is the first step in making sure that products/services are in line with market trends and requirements. Often, this involves discussions on new product development, pipeline of rollout, resource allocation for product lines or sun setting decisions. In the context of education leadership, the top leaders must ensure that programmes and courses are relevant to the industry. Specifically, a periodic review of programmes and curricula must be carried out to ensure issues such as industry relevance and trends are captured within the body of knowledge, courses and programme learning outcomes. A good practice is to run new programmes to be launched through several feasibility gates/checkpoints to ensure they are relevant.

### 6.3.2. Demand Review

A key aspect of this phase is for organisations to conduct proper demand forecasting using data from dependent and independent demand factors. The end goal is to develop a consensus-based demand plan, which is derived from market analysis and review. To ensure long-term sustainability, education institutions must remain relevant. The programmes or courses offered must be in line with industry trends and requirements. The number of jobless graduates in Malaysia is currently estimated at just over 170,000 people. This is largely due to universities that are churning out graduates that possess qualifications not required/irrelevant to industry trends. As such, thinking of programmes that are relevant to industries require proper demand review—universities must be fully aware of future skills and jobs that will be in demand.

### 6.3.3. Supply Review

An integral part of the S&OP process is to ensure that the business has the ability to offer what the market needs, that is, the ability to meet market demand. Typically, the demand and supply review process work in tandem with one another. Business simulations ('what-if' analyses) play a vital part in making sure the resource allocation is planned for accordingly in anticipation of demand. For an academic institution, this step requires that the organisation has sufficient talent, technology, resources such as labs/classrooms processes and overall capacity in light of its programme offering. For example, most universities are now investing in resources that mirror the fourth IR requirements—to ensure they remain relevant.

### 6.3.4. Finance Review

When a financial review should be done is debated. Some call for a review after the third step, but others say that financial review should be an ongoing process. From an S&OP process view, a financial review is often under the purview of the finance team—with a global business lens in mind—namely, looking at financial models and data, based on the customer, product, market and other data. Regardless of where this process sits in a company's overarching process, a financial review must be used as an input to pre-S&OP and executive S&OP steps. In the context of academic institutions, finance reviews need to take into consideration issues such as programme USP, relevance to industry and data analytics driven by past data (e.g., student numbers, employability rates, graduating on time requirements, progression estimates, etc.). When this is done, issues such as business sustainability can be better addressed.

### Pre-S&OP

Pre-S&OP usually involves a series of meetings between key leadership teams—often going through data and dashboards accordingly to assess gaps, variances, and, based on financial data, set out strategic and tactical plans. Alignment to product/service and supply plans are then done. For academic programmes, continuous improvements based on financial and market data are vital to ensure long-term sustainability of institutions. Pre-S&OP allows leadership within academic institutions to decide between growing versus canning of programs, rethink mode of delivery (supply) in light of market demand-driven data and very quickly rethink its new product development processes where necessary. Again, this will ensure that only programmes that are in demand and in tune with industry demands are offered and continuously improved.

Executive S&OP

The final stage of the S&OP process is the end game in mind. Decisions are made, and deadlines are set based on the feedback and input from the first five steps, and this applies to academic and other institutions alike. Academic leaderships that assume an S&OP process view for their respective institutions, are well placed to set themselves apart from the rest. The Future of Jobs 2020 (World Economic Forum) states that the top five skills that are highly demanded by industries in Malaysia are emotional intelligence, creativity, analytical thinking and innovation, technology design and programming and complex problem solving. The report also suggests that employers are focusing on similar skill sets as part of their corporate retraining and up skilling initiatives.

As such, to ensure they remain relevant, academic and training institutions alike need to offer programmes and courses that address these emerging skills that are required for the future. However, to reap the full benefits of the S&OP processes, a greater financial review and inclusion across the business and academic value chain is the way forward.

## 7. Discussion: Frugal Open Innovation

The concept of FOI plays an important role in business model, and it can be an alternative to introduce frugality in PHEI. FOI can be developed by redesigning business operations and processes to be more cost effective, open and functional. As presented in Table 10, all of the practitioners are aware and have heard about the term 'frugal' and 'open' concepts. The key concepts of FOI are value consciousness, spend less due to budget constraint, reduce wastage, minimum use of resources and external collaboration. Similarly, in Figure 9, a few terms can be spotted in word cloud as very frequently mentioned such as efficient, effective, value, cost and sustainable.

Notably, some changes in the innovation landscape are seen in many parts of the world as a result of revolution of information and digital technologies. Ref. [44] pinpoints a few expansions in digital technology which are spreading fast in education, health, farming, banking, insurance and many more fields to the developing world: 'artificial intelligence (AI), robotics, autonomous vehicles (including drones), the Internet of things (IoT), and 3D printing'. What is FOI in the digital era? ITC and IC are associated with FOI as a way forward to have cost efficiency instead of just cutting the cost by reducing resources, functionality and features of the products or services. As a result of reconfigurations of high technology components, firms can reduce complexity and achieve substantial cost savings. PHEI with a strong IC management and ITC able to satisfy the customer's requirement, thereby increasing the ability of firms to best utilise their resources consumption and produce more with less resources for cost efficiency. Based on this rationale, this research argues that IC and ITC play an important role to achieve FOI.

The investigation of IC and ITC through a FOI lens makes numerous features stand out. Firstly, FOI does not necessarily reuse the present technologies or resources but rather utilise more advanced technologies to develop resource-efficient and sustainable method with high socio-economic impact for firms. Therefore, emphasis must be given on exploring FOI principles for designing business solutions with less and efficient use of resources to achieve core functionalities and ideal performance level. Secondly, by developing IC and ITC, products and services can be produced frugally. Instead of using high expenditure on building hardware, it can be substituted by using software that incurs less fixed costs, stimulates new business and delivery models, reduces investments and increases productivity. Thirdly, there have been various opportunity for people to access, create and learn with electronics tools and 'software and digital fabrication'. These developments help academicians and researchers gain access to technologies that support them to embark more on innovation-driven research and collaboration with foreign or local institutions and industries as well. Hence, this research aims to find the effect of IC management and ITC that leads to FOI to generate higher performance.

**Table 10.** Awareness on frugal innovation for PHEI.

| Practitioners | Statement |
|---|---|
| Practitioner 1 | "Doing more with less has been there for a long time. Why waste if we can prevent it. It is actually being efficient over time. We should be able to do the same with less or more with less. Frugal innovation even it is not mentioned but in always inherent part of innovation. we want to make sure what we innovated adds more value than cost. It does not mean cheaper but must be value adding as well." |
| Practitioner 2 | "Frugal innovation is very timely and It is a new term for me. It is something that we do but we don't use that terminology. Frugal innovation maybe I can understand the concept by resource efficiency and doing more with less. Happy able to learn new concept." |
| Practitioner 3 | "Yes, heard on frugality concept which is being prudent in spending, thriftiness, don't waste money. But sometimes we need to spend more in order to provide better services for our students. It is very difficult to decide how much to spend and what to expect in terms of the return value. Universities need to educate more students with less resources used." |
| Practitioner 4 | "Not really, but if talking about cost reduction, minimum usage of resources and reduce wastage. Of course, I am familiar but not the concept of frugal innovation in depth." |

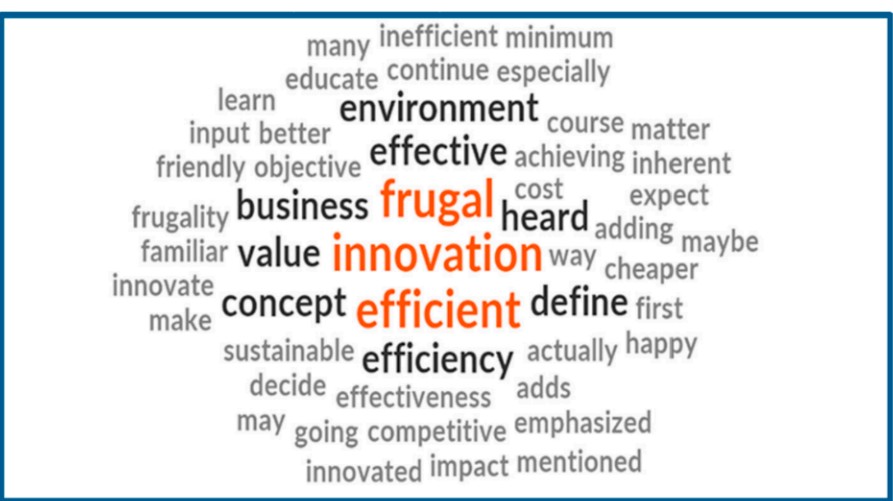

**Figure 9.** Word cloud diagram for frugal innovation.

## 8. Conclusions: Reforming Malaysian PHEI with FOI

FOI is viewed in terms of affordability, cost efficiency, value adding, simplicity and external collaboration, which are considered as the five main aspects of FOI. FOI aims to create value by developing products and services to address the right needs of the consumers and stakeholders through simpler, functional and affordable means at an adequate level. FOI, similar to other types of innovation, helps lower the cost by adopting creative business approaches in the business model. Our findings derived via analysis of a relatively small dataset, clearly suggest that no remedies and universal solutions can enhance the sustainability of PHEI unless they move on to frugal and open approaches to overcome the challenges and resource constraints to benefit wider society. Hence, PHEI are left with no choice but to strengthen their IC potential and ITC to survive in global competition, as summarised in Figure 10.

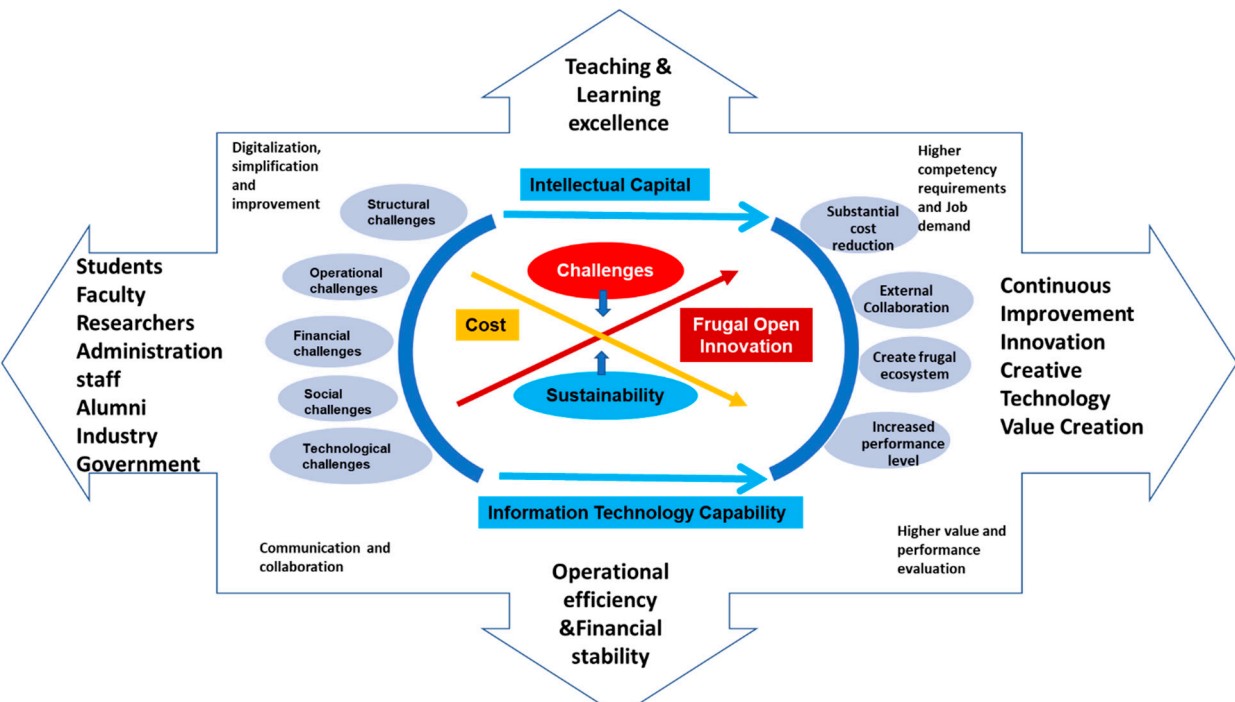

**Figure 10.** PHEI Framework for FOI.

As illustrated in Figure 10, in order to achieve FOI, IC and ITC must be improved. PHEI need to grasp opportunities and utilise their capabilities to ensure FOI, which leads to achievement of sustainability and reduced challenges. As a result, PHEI will be able to focus on core roles of PHEI; teaching and learning, operational efficiency and financial stability, continuous improvement, innovation, creative technology, value creation, and closer integration with internal (student, faculty, researchers, administration, and staff) and external stakeholders (industry, government, local and international communities).

There is lack of research that explores the relationships between organisational IC dimensions and ITC, to achieve FOI in the PHEI. Hence, the purpose of this study is to investigate the research gap by examining the effect of IC and ITC on FOI from the practitioner's view. Greater understanding is required on resource use (how it improves research, learning and teaching), business practice and equal distribution (of the benefits of stakeholders across different societies).

The findings of this study can be of use to researchers' academic communities in emerging economies in general. Our discussion on FOI and its relationship to IC, ITC, and the S&OP ideas, can be a starting point for PHIEs in emerging economies that are subject to similar challenges that is faced in Malaysia. Strategies that foster the use of FOI concepts can then be explored accordingly.

This study is expected to contribute to create continuously evolving approaches of IC knowledge to develop effective IC management through various other capabilities. This research proposes to shed light on the ability of PHEI to formalise, capture and leverage their intangible assets rather than only investing and managing tangible assets to attain FOI. In this regard, there is a scope for more work to be done on the role of IC and ITC as well as how this elements influence PHEI to achieve FOI. Hence, this paper is timely in informing scholars within the realm of IC research especially in PHEI to fill this gap.

**Author Contributions:** Conceptualization, J.J., M.D. and M.R.; methodology, J.J. and M.D.; validation, J.J., M.D. and M.R.; formal analysis, J.J., M.D. and M.R.; investigation, J.J.; resources, J.J. and M.D.; data curation, J.J., M.D. and M.R.; writing—J.J.; writing—review and editing, J.J., M.D. and M.R.; visualization, J.J. and M.D.; supervision, M.D. and M.R.; project administration, M.D.; funding acquisition, M.D. All authors have read and agreed to the published version of the manuscript.

**Funding:** This research was funded by the Ministry of Higher Education, Malaysia under the Fundamental Research Grant Fund (FRGS/1/2020/SS02/MMU/02/3).

**Institutional Review Board Statement:** Research Ethical Committee (REC) of Multimedia University (EA0112021). The study was conducted according to the guidelines and approved by the Research Ethical Committee (REC) of MULTIMEDIA UNIVERSITY.

**Informed Consent Statement:** Informed consent was obtained from all subjects involved in the study.

**Data Availability Statement:** All data is provided in this paper.

**Acknowledgments:** We thank the Ministry of Higher Education, Malaysia for awarding a Fundamental Research Grant Fund (FRGS/1/2020/SS02/MMU/02/3) to conduct this research. The findings shared in this paper is part of this project.

**Conflicts of Interest:** The authors declare no conflict of interest. The funders had no role in the design of the study; in the collection, analyses, or interpretation of data; in the writing of the manuscript, or in the decision to publish the results.

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
