# Peer review of "Reshaping Higher Educational Institutions through Frugal Open Innovation"

_2199-8531, doi:10.3390/joitmc7020145_

Round 1

Reviewer 1 Report

The paper claims to "fill the gap in the existing literature on HLIs’ performance measurement initiatives, underlining the growing importance of IC, which is often overlooked in the academic discourse and by policy makers and marketers."

And the authors claim that "there is a growing need for HLIs to improve their efficiency and effectiveness" when what their data shows that they lack financing.

In my modest opinion, the paper is fat too much biased towards "stakeholders’ interests rather than focusing in the purpose of education and the interest of students and society. I recommend them to read Dewey. 

References are not numbered by order of appearance.

Why they only focus in private institutions?

The explanation for HLI comes after using it but it should appear at first use.

Why do authors use "Higher learning institutions" instead of "Higher education institutions" students learn but institutions teach/educate. 

In the introduction there is a mixture of normative, evaluative and descriptions, this is really confusing.

Is Relational capital what is mostly known as "Social capital" there is a lack of review on this literature.

Only 4 interviews were carried out, seems to little to get a perspective.

This paragraph The above themes can be categorized into four quadrants of challenges. The preceding section looks at technology as a standalone challenge, it can also be grouped, in our view, alongside operations as technology and the HLI information systems in general. As such, technological challenges are pervasive and can cut across the value chain." is repeated twice.

The part of the interviews is disconnected from section 6. It does not seem like a coherent research but a disconnected compilation of assays.

Author Response

Dear Respected Reviewer,

We have compiled the suggestions and comments as per your call, and have addressed the concerns based on your excellent suggestion. Please find the attached response to review document for details. 

Thank you

Reviewer 2 Report

  1. This paper does a content analysis of the opinions from 4 practitioners on reshaping higher education. The current version clearly presents it research process and results.  However, some fundamental questions remain to clarify or improve.
  2. Figure 1 is a conceptual framework with a structural form, in which capabilities mediate between three types of capital and frugal open innovation. However, in the content analysis, it is not clear how capabilities mediate between three types of capital and frugal open innovation.  That is, the content analysis should be consistent with the conceptual framework depicted in Figure 1.
  3. Actually the conceptual framework can be statistically verified by questionnaires and structural equation modeling (SEM). It is not clear why the standard survey and statistical analysis is not followed and a content analysis of 4 practitioners’ opinion is sufficient to verify the conceptual framework depicted in Figure 1.
  4. The word cloud diagrams used in this paper are usually applied to summarize the most frequent key words crawled on Internet. Here there are only opinions from four practitioners.  It is then very far away from the big data results as most world cloud diagrams do.  It is not clear why not more practitioners were invited to give opinions such that a larger sample size is used to generate the word cloud diagrams.
  5. The verbal opinions summarized in tables are direct effects on re-shaping higher education. There is no meditation effect of capabilities verified or presented in these tables at all.  If the authors want to verify the conceptual framework depicted in Figure 1 by content analysis of 4 practitioners, the research steps should be corresponding to the conceptual framework depicted in this paper.  However, the current writing is verifying another framework (or no framework).
  6. The academic contributions obtained from summarizing these 4 practitioners should be further elaborated and highlighted. For instance, the new academic propositions or conceptual frameworks obtained from the wisdom of these 4 practitioners should be explicitly presented.

Author Response

(The authors gave the same response as above.)

Reviewer 3 Report

The topic is relevant, as its results can contribute to attracting and retaining students in higher education, and consequently offer a competitive advantage. However, the financial results of these types of institutions depend to a large extent on factors external to the institution itself, which is why it chooses the market segment with well-defined criteria through differentiating educational products and appropriate technological pedagogical tools, such as teaching in e-learning mode. Research centers and their relationship with industry can be an excellent source of revenue. The relationship between private universities and private industry can represent a competitive advantage, given that in many situations administrative processes can be faster

In the last paragraphs of the Abstract, authors are suggested to identify the main results of the investigation, namely to highlight the factors or best practices among the variables under study.

In the Introduction, the authors provide a sufficient framework, however it is not clear how information technologies can represent a competitive advantage for private higher education. The authors do not justify why the study was carried out in the private sector. Intellectual capital and innovation depend significantly on the leadership style of the different levels of the Institution, the organizational climate and the evaluation criteria of teachers, and these dimensions have not been addressed. At the end of the Introductions, authors are asked to formulate a research question to guide the study.

The literature review on the keywords is sufficient, however, it would have been interesting for the authors to have introduced other approaches, such as sustainability, social responsibility, industry 4.0, digital context, lean management and circular economy. Literature review is not enough as to the relationship between private Higher Learning Institutions, industry and the local community. It would also have been interesting for the authors to have carried out a literature review on the relationship between university research centers - industry - prototypes. At the end of the literature review and within the framework of the theoretical framework, authors are asked to ask research questions.

In the subsection methods, it is considered that 4 interviews are insufficient to respond to the challenges of this investigation. The authors do not explain how they do the content analysis of the respondents' responses. Content analysis of respondents' responses does not follow traditional methods.

Suggestions for authors:

  • Formulate a starting research question.
  • Since the qualitative method was used in the investigation, identify research questions.
  • Focus on the text of the study according to the theme of the paper, namely to frame it with the word “private”.
  • Conduct more interviews until the “saturation point” is obtained.

Author Response

(The authors gave the same response as above.)

Round 2

Reviewer 1 Report

The paper has improved in the revision process.

Author Response

Dear Respective Reviewer,

We thank you for the opportunity to revise the paper and value each of your comments. We trust that the paper reads better now. Hope to receive a positive outcome.

Thank you.

Reviewer 2 Report

  1. As the revised manuscript shows, the authors have put efforts in revising this paper. The authors also removed all the ‘mediation effect’ terms from the abstract to the conclusion since the first copy actually does not analyze it.  The key sentences such as “exploring the effect of intellectual capital on frugal innovation is mediated through the information technology capability” in previous version simply disappeared.  Some minor but important problems remain to clarify or improve.
  2. The conceptual framework in Fig. still contains mediation effect. Moreover, it is a full mediation effect.  Since the authors kept saying that they were not analyzing the mediation effect at all, the mediation effect in the conceptual framework in Fig. 1 should be removed.  This paper has only direct effects as Fig. 10 indicates.  All the related statements in the context should be expressed as direct effects.
  3. Table 8 presents now clearly the key results of content analysis. It should be further emphasized and interpreted.  However, Table 8 seem to have some typos:  vis-[à-]vis, compensation[,]etc.  Careful English proofreading and editing is necessary.
  4. This paper can be an illustration of content analysis of small data. The value of small-data content analysis should be further highlighted.
  5. In the conclusion application the proposed extended framework of this paper to students in emerging economies can be further emphasized.

Author Response

(The authors gave the same response as above.)
